# Photoluminescent Lanthanide(III) Coordination Polymers with Bis(1,2,4-Triazol-1-yl)Methane Linker

Elizaveta A. Ivanova [1], Ksenia S. Smirnova [1], Ivan P. Pozdnyakov [2], Andrei S. Potapov [1,*] and Elizaveta V. Lider [1,*]

1   Nikolaev Institute of Inorganic Chemistry, Siberian Branch of the Russian Academy of Sciences, 630090 Novosibirsk, Russia; ivanovaea@niic.nsc.ru (E.A.I.); smirnova_ksenya96@mail.ru (K.S.S.)
2   Voevodsky Institute of Chemical Kinetics and Combustion, Siberian Branch of the Russian Academy of Sciences, 630090 Novosibirsk, Russia; pozdnyak@kinetics.nsc.ru
*   Correspondence: potapov@niic.nsc.ru (A.S.P.); lider@ngs.ru (E.V.L.)

**Abstract:** A series of new lanthanide(III) coordination polymers with the general formula $[Ln(btrm)_2(NO_3)_3]_n$, where btrm = bis(1,2,4-triazol-1-yl)methane and Ln = $Eu^{3+}$, $Tb^{3+}$, $Sm^{3+}$, $Dy^{3+}$, $Gd^{3+}$ were synthesized and characterized by IR-spectroscopy, elemental, thermogravimetric, single-crystal, and powder X-ray diffraction analyses. Europium(III), samarium(III), terbium(III), and gadolinium(III) coordination polymers demonstrate thermal stability up to 250 °C, while dysprosium(III) is stable up to 275 °C. According to single-crystal X-ray diffraction analysis, the ligand exhibits a bidentate-bridging coordination mode, forming a polymeric chain of octagonal metallocycles. The photoluminescence of the free ligand in the polycrystalline state is observed in the ultraviolet range with a quantum yield of 13%. The energy transfer from the ligand to the lanthanide ions was not observed for all obtained coordination polymers. However, there are sharp bands of lanthanide(III) ions in the diffuse reflectance and excitation spectra of the obtained compounds. Therefore, Ln(III) luminescence arises, most probably, from the enhancement of f-f transition intensity under the influence of the ligand field and non-centrosymmetric interactions.

**Keywords:** europium; terbium; samarium; dysprosium; gadolinium; bis(1,2,4-triazol-1-yl)methane; coordination polymer; photoluminescence; crystal structure

## 1. Introduction

The synthesis and investigation of novel rare earth elements (REE) coordination compounds is a promising direction in modern coordination chemistry because of the attractive luminescent properties of these compounds, since lanthanide ions can emit light both in the visible and near-infrared ranges. The organic ligands in lanthanide complexes perform a key role, as far as REE luminescence mainly occurs due to the energy transfer from the excited state of the ligand to the emitting state of lanthanide(III) cation ($Ln^{3+}$). Currently, this is considered to be the main way to improve the emission and is called the antenna effect [1–3]. Luminescence due to 4f-transitions is of special interest since 4f-orbitals are shielded from 5s and 5p ones. As a result, the emission bands are quite sharp, while the luminescence is pure in color and individual for REE [4–6]; for example, it is red for the europium(III) [7,8] and green for the terbium(III) ion [9,10]. Samarium(III) and dysprosium(III) complexes receive much less attention since their luminescence intensity is usually low compared to europium(III) and terbium(III) complexes. However, dysprosium(III) complexes can be used to obtain not only luminescent but also magnetic materials. Tong and coworkers synthesized several dysprosium(III) complexes and investigated their magnetic properties. According to the authors, the properties of single-molecule magnets can be significantly improved by optimizing the ligand field. The decrease in electrostatic repulsion between the ligands and f-electrons enhances the uniaxial anisotropy [11].

1,2,4-Triazoles are five-membered heterocycles with three nitrogen donor atoms and are widespread as ligands in transition metals coordination compounds [12]. Owing to various possible coordination modes, the ligands based on 1,2,4-triazoles give rise to interesting coordination chemistry, in particular, for the design of coordination polymers and metal–organic frameworks. For the protonated and deprotonated forms, the 1,2,4-triazole ring can exhibit bidentate coordination mode via nitrogen atoms, as well as tridentate by nitrogen atoms upon deprotonation [13]. The interaction of REEs with 1,2,4-triazole usually leads to three-dimensional frameworks. However, Müller-Buschbaum et al. obtained a two-dimensional polymer structure by reacting metallic holmium with melted 1,2,4-1H-triazole. The coordination number of the holmium ion is equal to nine, and the coordination polyhedron is close to the pentagonal bipyramid. Five deprotonated and two neutral triazole molecules participate in coordination. In this case, two triazolate ions exhibit bidentate-bridging coordination mode by nitrogen atoms, while the remaining three ones, as well as the neutral triazole molecules, demonstrate monodentate coordination via the nitrogen atoms [14].

1,2,4-Triazole derivatives are known to possess diverse pharmacological properties, such as antitubercular [15], anticancer [16,17], antifungal [18,19], antitumor [20,21], and antibacterial [22,23] activities. 1,2,4-Triazole fragments in these compounds are responsible for non-covalent interactions with the circulating enzymes and the receptors in organs; consequently, such compounds are able to improve the solubility and can bind to bimolecular targets [24,25]. Moreover, 1,2,4-triazole derivatives often exhibit bright blue fluorescence with tunable electronic and physical properties favorable for blue OLED emission [26,27].

Triazole derivatives can be divided into two groups: symmetric and asymmetric ligands [28–30], which act as chelate ligands. For instance, Yang et al. used the 6-(1H-1,2,4-triazol-1-yl)pyridine-2-carboxylate and oxalate as ligands to synthesize new isostructural mixed-ligand lanthanide(III) polymeric compounds, namely europium(III), terbium(III) and gadolinium(III), which demonstrate high quantum yields. The terbium(III) complex was found to respond effectively to biomarkers of carcinoid cells, while the europium(III) compound effectively distinguished quercetin. In addition, the authors developed a photostable luminescent ink based on the obtained complexes [28]. Bis(1,2,4-triazol-1-yl)pyridines are among the symmetrical ligands with two 1,2,4-triazole rings, the coordination chemistry of which, with the lanthanides(III), is well explored in the literature [31,32]. Some of these compounds demonstrate bright photoluminescence and high solubility in organic solvent, which is important for bioimaging and fabrication of electroluminescent devices [33–35]. Another example of a symmetrical ligand is bis(pyridin-2-yl)-1,2,4-trizole-1-yl, which was applied by Guo et al. for the synthesis of a tetranuclear dysprosium(III) complex. The obtained compound exhibited white light emission by changing the excitation wavelength. This was achieved by the dichromatic emission mixing of the ligand (blue) and the dysprosium ion (yellow) [36].

The coordination chemistry of bis(1,2,4-triazol-1-yl)methane (btrm, Scheme 1) is much less explored. Despite the fact that coordination polymers and metal–organic frameworks based on btrm and transition metal cations showing luminescent and gas adsorption properties were reported [37–42], there are no examples of lanthanide coordination compounds based on btrm or similar ligands in the literature.

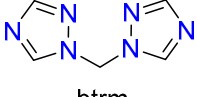

btrm

**Scheme 1.** Bis(1,2,4-triazol-1-yl)methane (btrm) ligand used in this work.

In this work, we report the synthesis, structural characterization, and luminescent properties of the first examples of lanthanide(III) coordination polymers with bis(1,2,4-triazol-1-yl)methane as a linker.

## 2. Results and Discussion

### 2.1. Synthesis of the Coordination Polymers

Lanthanide(III) coordination polymers with the general formula $\{[Ln(btrm)_2(NO_3)_3] \cdot xH_2O \cdot yMeCN\}_n$ (**1–5**) were synthesized by mixing the acetonitrile solutions of the btrm ligand and lanthanide(III) nitrates in 2:1 molar ratio followed by heating at 60 °C (Scheme 2). The products were separated as microcrystalline precipitates after evaporation of about half of the solvent from the reaction mixture. The elemental analysis data were consistent with the proposed formula of the compounds (Scheme 2).

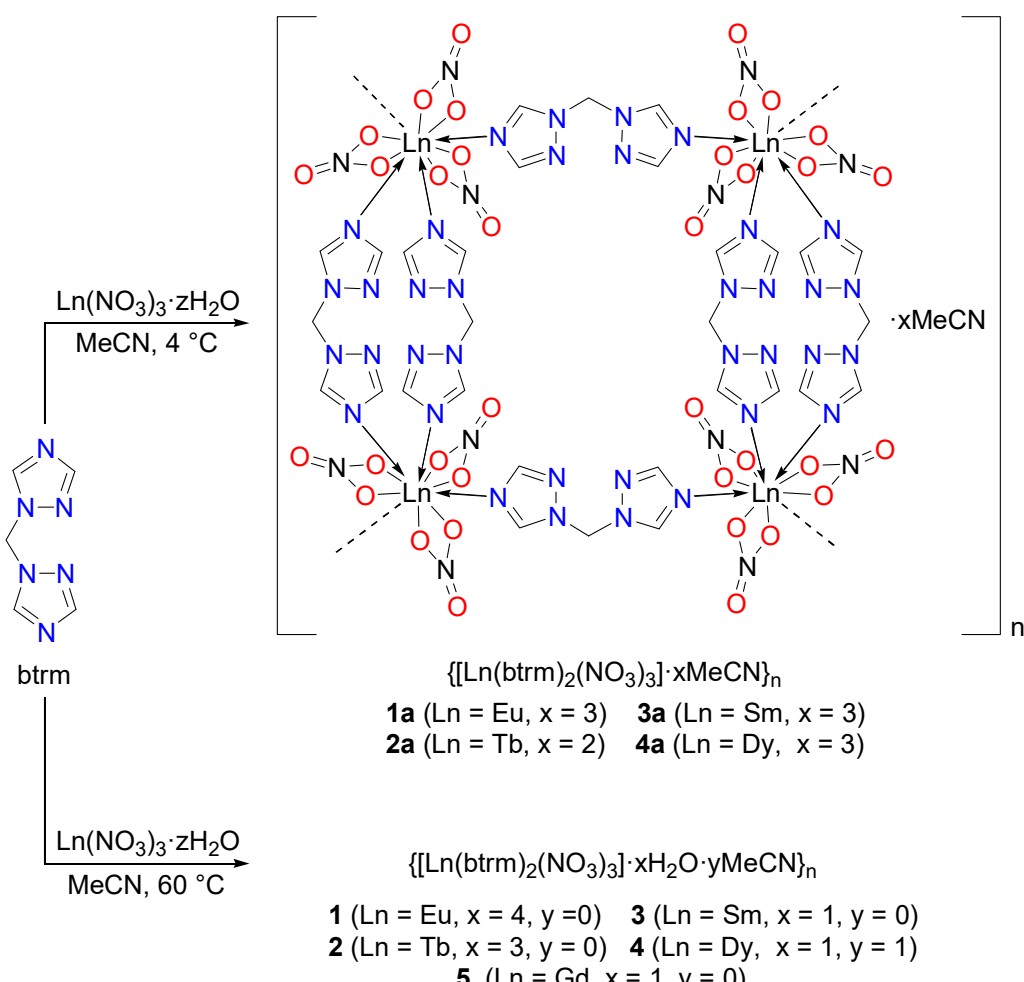

$\{[Ln(btrm)_2(NO_3)_3] \cdot xMeCN\}_n$

**1a** (Ln = Eu, x = 3)    **3a** (Ln = Sm, x = 3)
**2a** (Ln = Tb, x = 2)    **4a** (Ln = Dy, x = 3)

$\{[Ln(btrm)_2(NO_3)_3] \cdot xH_2O \cdot yMeCN\}_n$

**1** (Ln = Eu, x = 4, y = 0)    **3** (Ln = Sm, x = 1, y = 0)
**2** (Ln = Tb, x = 3, y = 0)    **4** (Ln = Dy,  x = 1, y = 1)
**5** (Ln = Gd, x = 1, y = 0)

**Scheme 2.** Synthesis of lanthanide(III) coordination polymers.

### 2.2. Crystal Structures of the Coordination Polymers

No single crystals could be obtained from the mother liquors. Single crystals of compounds $\{[Ln(btrm)_2(NO_3)_3] \cdot xMeCN\}_n$ (**1a–4a**) were obtained upon cooling the reaction mixtures to 4 °C immediately after mixing the btrm and $Ln(NO_3)_3$ solutions (Scheme 2).

The coordination compounds $\{[Ln(btrm)_2(NO_3)_3] \cdot xMeCN\}_n$ (**1a–4a**) are isostructural, crystallize in a monoclinic crystal system, space group $P2_1/n$ for Ln = $Eu^{3+}$, $Sm^{3+}$, $Dy^{3+}$, and space group $C2/m$ for Ln = $Tb^{3+}$. Substructures with the $C2/m$ space group were additionally found for europium(III) and dysprosium(III) compounds (**1b** and **4b**). Crystal structures with $P2_1/n$ space group featured a glide plane passing through the $CH_2$ group of btrm (Figure S1a), while for the structures with the $C2/m$ space group, there were the mirror and glide planes, as well as a two-fold axis passing through the $Ln^{3+}$ cation and the N–O bond of one of the nitrate anions (Figure S1b).

The coordination sphere of the lanthanide cation in {[Ln(btrm)$_2$(NO$_3$)$_3$]·xMeCN}$_n$ compounds consists of six oxygen atoms of three bidentate nitrate anions and four nitrogen atoms of four bidentate bridging btrm ligands, which join the Ln$^{3+}$ cations into a polymeric chain. The monomeric unit of this chain may be described as octagonal metallocycles [Ln$_4$L$_4$(NO$_3$)$_{12}$], containing four lanthanide(III) cations and four btrm molecules (Figure 1a). The Ln$^{3+}$ cations are bridged by one btrm molecule along the crystallographic axis *c* and by two btrm molecules along the axis b. As a result, a chain of interlocking octagons oriented along the crystallographic axis *c* is formed (Figure 1b). The acetonitrile solvate molecules occupy the space between the chains of the coordination polymer (Figure 2). In the case of europium(III), samarium(III), and dysprosium(III) compounds, the number of MeCN in [LnL$_2$(NO$_3$)$_3$] moiety is equal to three, while for the terbium(III), it is two. The intermolecular contacts C–H⋯O between the neighboring C–H groups of 1,2,4-triazole rings or MeCN with the oxygen atom of the nitrate group results in the packing of the chains into a supramolecular structure (Figure 3). According to the continuous shape measure analysis [34], the coordination polyhedron can be described as a sphenocorona with S(C$_{2v}$) = 3.66, 3.87, 3.71, and 3.58 for **1a–4a**, respectively, or a bicapped square antiprism with S(D$_{4d}$) = 3.44, 3.29, 3.47, and 3.23. The bond lengths Ln–N and Ln–O in the structure of compounds **1a–4a** are presented in Table 1.

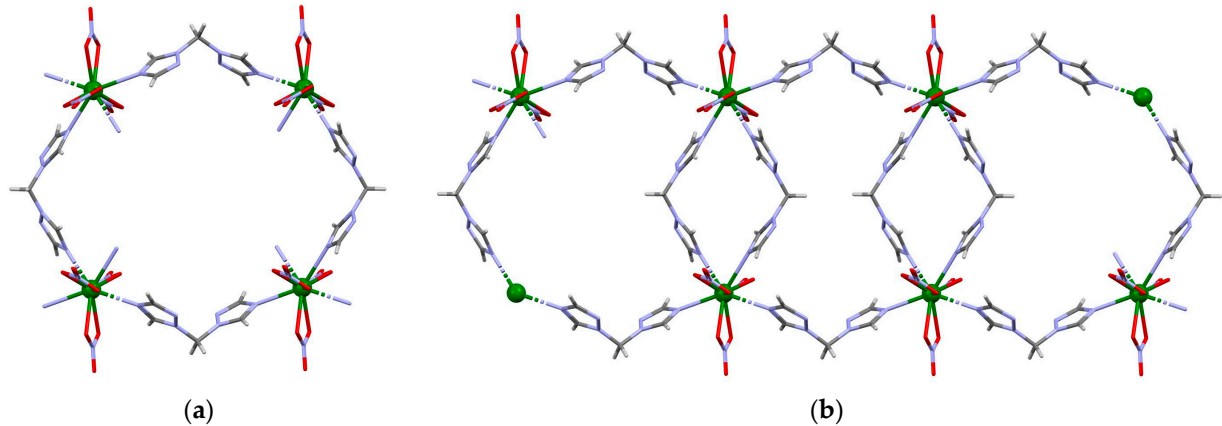

(**a**)                                                    (**b**)

**Figure 1.** Crystal structure of the coordination polymers {[Ln(btrm)$_2$(NO$_3$)$_3$]·xMeCN}$_n$ (**1a–4a**), solvent molecules are omitted for clarity: (**a**) monomeric unit; (**b**) chain of coordination polymer oriented along the crystallographic axis *c*.

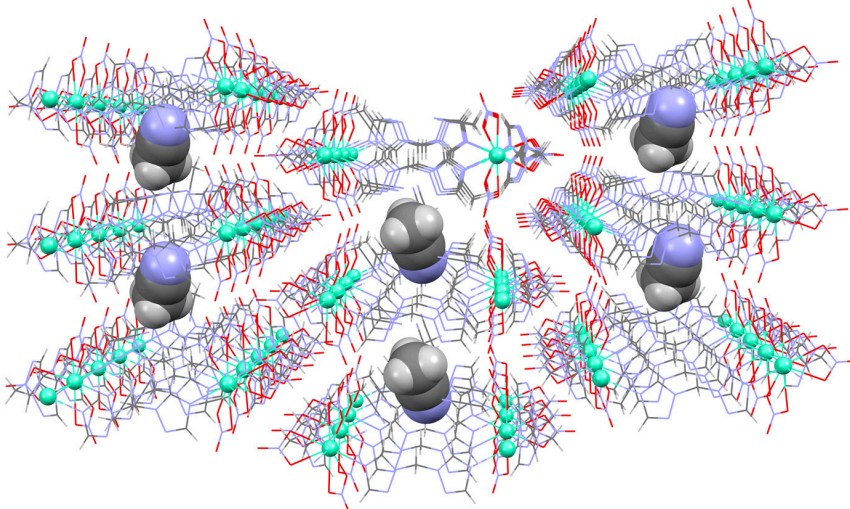

**Figure 2.** The location of MeCN molecules in the structure of the coordination polymers {[Ln(btrm)$_2$(NO$_3$)$_3$]·xMeCN}$_n$ (**1a–4a**).

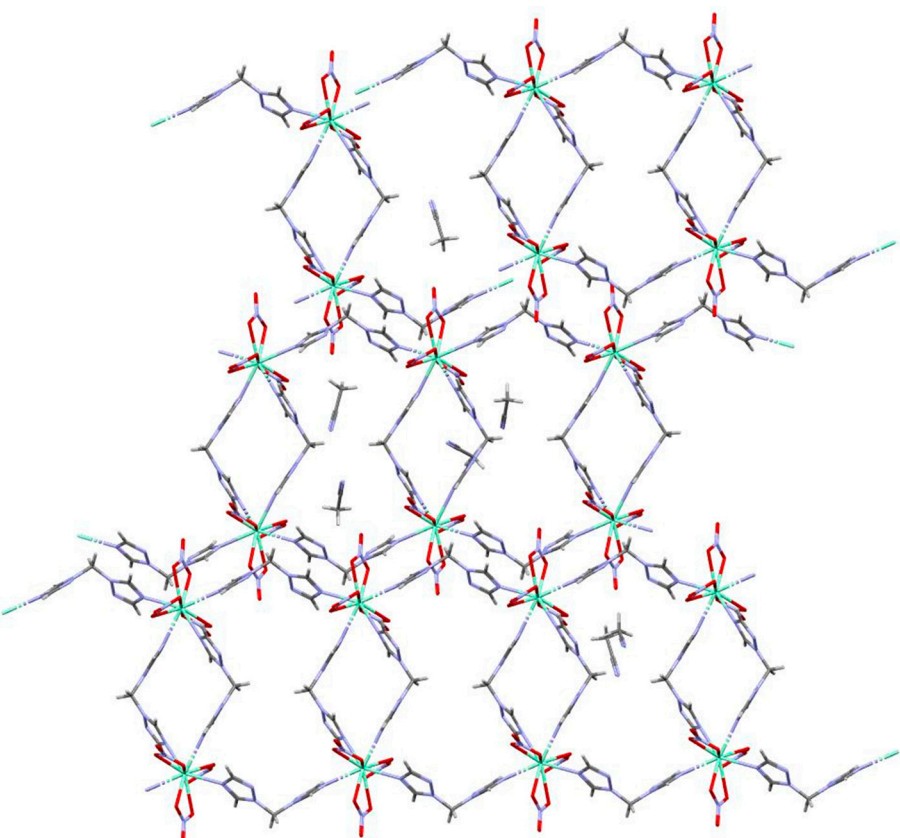

**Figure 3.** The packing of $\{[Ln(btrm)_2(NO_3)_3]\cdot xMeCN\}_n$ chains.

**Table 1.** The bond lengths (Ln–N and Ln–O) in the structure of the coordination polymers **1a–4a**.

| Compound | d(M–N), Å | d(M–O), Å |
|---|---|---|
| **1a** | 2.578(3) 2.579(3) 2.600(4) 2.587(4) | 2.529(3) 2.484(3) 2.499(3) 2.575(3) 2.466(3) 2.536(3) |
| **2a** | 2.544(5) 2.564(6) | 2.442(5) 2.483(4) 2.554(5) |
| **3a** | 2.596(3) 2.588(3) 2.601(4) 2.596(4) | 2.513(3) 2.536(3) 2.498(3) 2.486(3) 2.580(3) 2.543(3) |
| **4a** | 2.545(5) 2.542(4) 2.544(5) 2.561(5) | 2.421(4) 2.442(4) 2.494(4) 2.557(5) 2.458(4) 2.521(5) |

### 2.3. Thermogravimetric, Powder X-ray Diffraction Analyses, and IR Spectroscopy

According to thermogravimetric analysis, compounds **1–4** contain water molecules. Europium(III), samarium(III), terbium(III), and gadolinium(III) coordination polymers

decomposed at 250 °C, while the dysprosium(III) compound **4** was at 275 °C. (Figures S2–S5). The IR spectra of the coordination polymers were in agreement with their structure (Table S1). The bands of C–H stretching vibrations were located in the range of 3110–2850 cm$^{-1}$ and the bands of bending vibrations were at 1460–1380 and 1200–1020 cm$^{-1}$. The stretching vibrations of the nitrate anions were observed near 1490 cm$^{-1}$ and 1305 cm$^{-1}$. The separation between these bands of about 185 cm$^{-1}$ indicates a bidentate coordination mode of the nitrate anions. Wide bands in the region of ~3360 cm$^{-1}$ in the IR spectra of all coordination polymers were associated with the OH stretching vibrations of the solvate water molecules.

According to powder X-ray diffraction analysis, compounds **1a–4a**, the crystals obtained by cooling the reaction mixtures were isostructural. However, the polycrystalline products **1–5** resembled another isostructural series different from compounds **1a–4a** (Figure 4a), except for the dysprosium(III) compound {[Dy(btrm)$_2$(NO$_3$)$_3$]·H$_2$O·MeCN}$_n$ (**4**), which was isostructural to compound {[Dy(btrm)$_2$(NO$_3$)$_3$]·3MeCN}$_n$ (**4a**), according to PXRD analysis (Figure 4b).

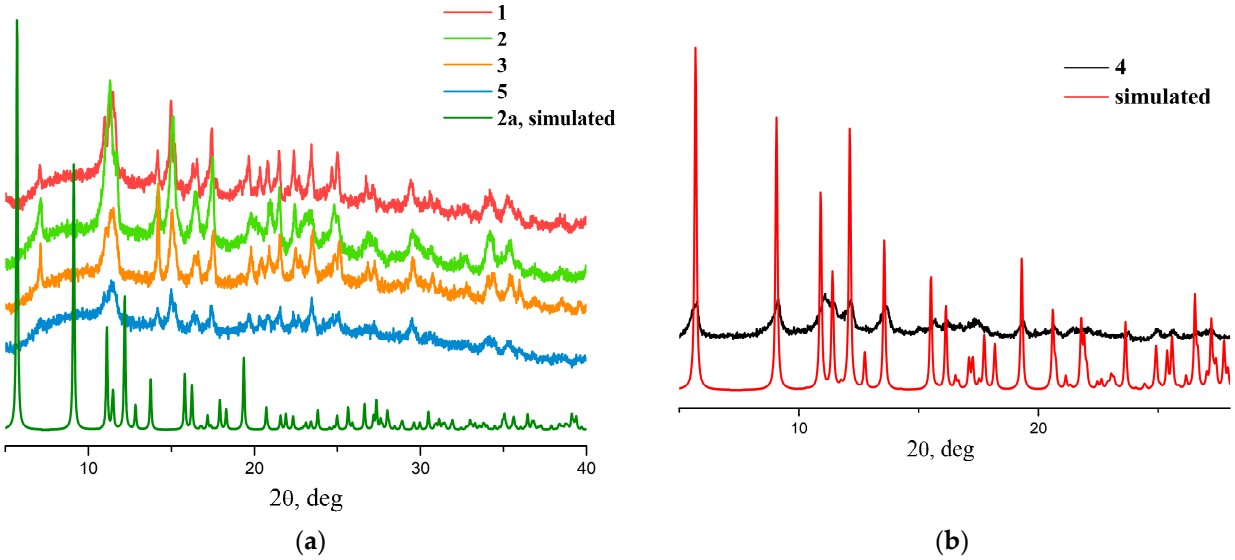

**Figure 4.** Experimental and simulated powder X-ray diffraction patterns: (**a**) for the coordination polymers **1–3**; (**b**) for the coordination polymer {[Dy(btrm)$_2$(NO$_3$)$_3$]·H$_2$O·MeCN}$_n$ (**4**).

It was found that the hydrated compounds **1–4** underwent solvent exchange upon keeping them in acetonitrile vapor for five days and transformed into acetonitrile solvates **1a–4a**, which was confirmed by PXRD analysis (Figure 5). After drying at room temperature, the compounds lose the acetonitrile molecules and return to their original state, which was also confirmed by PXRD data (Figure 5).

### 2.4. Photoluminescent Properties of the Coordination Polymers

The photoluminescent properties of the synthesized coordination polymers and btrm ligand were investigated for polycrystalline samples at room temperature. The btrm ligand exhibited a single-band fluorescence with a maximum of 325 nm upon excitation at 280 nm, the quantum yield was 13% (Figure 6). The photoluminescence decay kinetics of btrm is described by a monoexponential equation $I = A \cdot exp(-t/\tau)$ with the observed lifetime $\tau$ = 1.1 ns (Figure S6).

The diffuse reflectance, emission, and excitation spectra of the coordination polymers **1–4** are shown in Figure 6. Characteristic sharp emission bands of Ln(III) ions were detected in the luminescence spectra of all compounds. However, the excitation spectra of all coordination polymers do not contain characteristic bands of the ligand centered near 280 nm, though this band is clearly seen in diffuse reflectance spectra (Figure 6). In addition,

intensive and sharp bands specific for lanthanide(III) ions transitions were observed in the excitation spectra of all coordination polymers (Figure 6). The same bands, only less intense, can be found in the diffuse reflectance spectra, as shown for samarium(III) compound **3** (Figure 6).

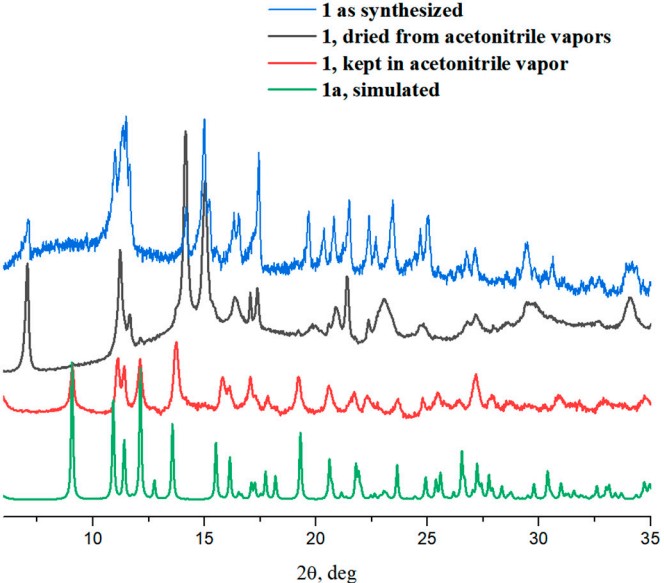

**Figure 5.** Experimental and simulated powder X-ray diffraction patterns for compound **1** before and after exchange of solvent molecules.

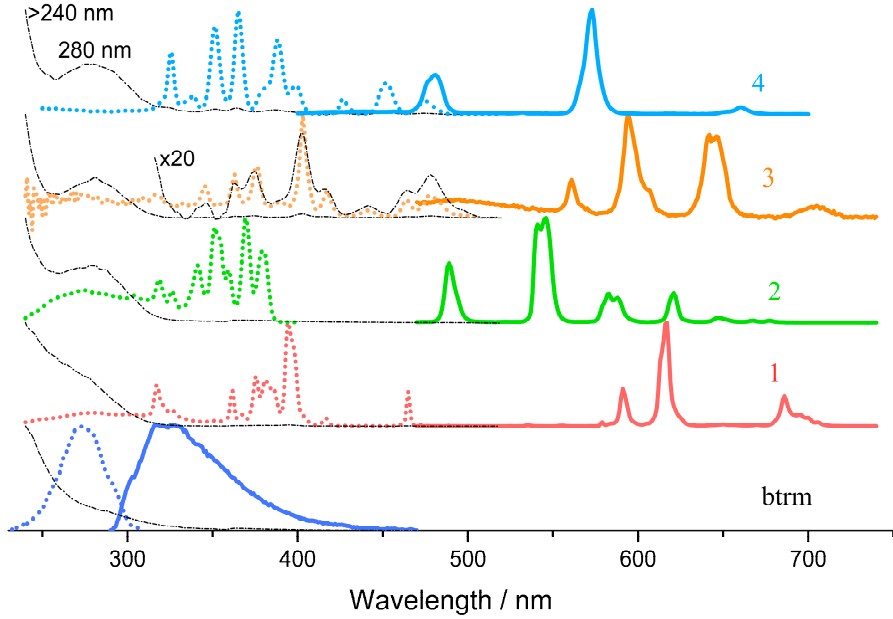

**Figure 6.** Diffuse reflectance (dotted black curves), excitation (dotted colored lines), and emission spectra (bold colored lines) for the ligand (btrm) and lanthanide coordination polymers **1–4**. For compound **3**, part of the diffuse reflectance spectrum is multiplied by a factor of 20 for better comparison with the excitation spectrum.

To clarify the situation, the phosphorescence of the ligand at low temperature (77 K) in the Gd(III) coordination polymer **5** was measured (Figure 7) and the position of the triplet state was estimated as 20,400 cm$^{-1}$ (490 nm). This meaning is also in agreement with the value predicted by the DFT-calculations, which is equal to 21,400 cm$^{-1}$. Therefore, in principal, good energy transfer should be observed in the case of Sm and Eu polymers, and no

transfer is possible for Tb and Dy complexes (the energy gap for compounds **1–4** calculated according to [43] is about 3200, 0, 2400, and $-700$ cm$^{-1}$, correspondingly). However, it can be suggested that there is no antenna effect since the characteristic absorption band of the ligand with the maximum at 280 nm was absent or only weakly present in comparison to the strong f-f transitions in the excitation spectra of all compounds (Figure 6).

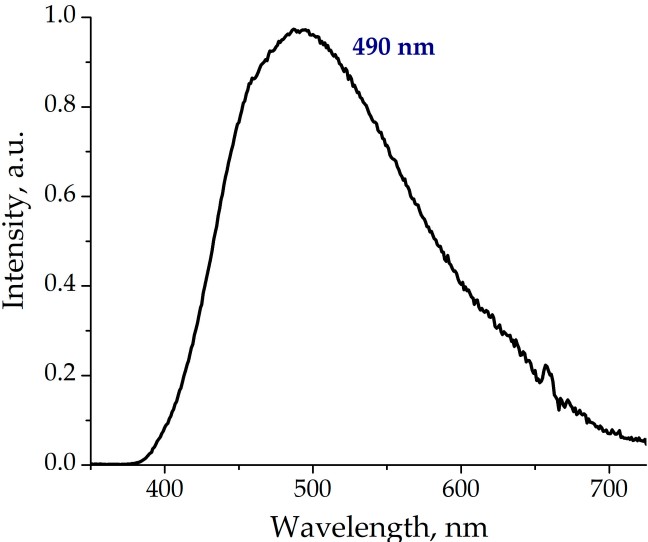

**Figure 7.** The phosphorescence spectrum of coordination polymer **5**, $\lambda_{ex}$ = 330 nm, 77 K, flash delay 0.15 ms.

Therefore, the energy transfer, common for Ln(III) luminescence sensitization [5] was not effective for these complexes. Most probably, the observed emission arises from the enhancement of f-f transition intensity via non-centrosymmetric interactions and possible mixing with appeared allowed transitions due to the presence of ligands (for example, LMCT transitions). As a result, the observed luminescence quantum yields upon excitation to the most intensive f-f transition were low and did not exceed 4% (Table 2). The luminescence lifetimes were typical for the corresponding lanthanides(III) (Table 2, Figures S7–S10).

**Table 2.** Luminescence lifetimes and observed quantum yields (QY) of the coordination polymers.

| Compound | Lifetime, ms | QY, % |
|:---:|:---:|:---:|
| **1** | 0.71 | 3 ($\lambda_{ex}$ = 395 nm) |
| **2** | 1.17 | 4 ($\lambda_{ex}$ = 370 nm) |
| **3** | 0.3 | 0.1 ($\lambda_{ex}$ = 403 nm) |
| **4** | 0.026 | 0.4 ($\lambda_{ex}$ = 365 nm) |

There is only a tentative explanation for the energy transfer absence in the case of obtained coordination polymers. Probably, an effective electron transfer from the excited ligand to the Ln(III) ion took place instead of an energy transfer leading to the absence of a common antenna effect.

### 3. Materials and Methods

*3.1. Synthetic Procedures*

All commercial reagents (lanthanide(III) nitrates) were purchased from Merck (St. Louis, MO, USA) and used as received. Bis(1,2,4-triazol-1-yl)methane (btrm) was synthesized as described earlier [44].

### 3.1.1. Synthesis of {[Eu(btrm)$_2$(NO$_3$)$_3$]·4H$_2$O}$_n$ (**1**) Coordination Polymer

An acetonitrile solution (1.0 mL) of btrm (0.40 mmol, 0.075 g) was added to an acetonitrile solution (1.0 mL) of Eu(NO$_3$)$_3$·6H$_2$O (0.20 mmol, 0.089 g) while stirring on a magnetic hot plate. The resulting solution was evaporated to about half of the initial volume at 60 °C, then cooled to room temperature. As a result, a white precipitate formed, which was filtered off, washed with acetonitrile, and air-dried. Yield: 76% (0.096 g). Elemental analysis (%): Calc. for C$_{10}$H$_{20}$N$_{15}$O$_{13}$Eu: C 16.9; H 2.8; N 29.5. Found: C 16.8; H 2.5; N 28.9.

### 3.1.2. Synthesis of {[Tb(btrm)$_2$(NO$_3$)$_3$]·3H$_2$O}$_n$ (**2**) Coordination Polymer

Compound **2$^{Tb}$** was synthesized similarly to compound **1** using Tb(NO$_3$)$_3$·5H$_2$O (0.10 mmol, 0.044 g) and btrm (0.20 mmol, 0.031 g). Yield: 86% (0.055 g). Elemental analysis (%): Calc. for C$_{10}$H$_{18}$N$_{15}$O$_{12}$Tb: C 17.1; H 2.6; N 30.0. Found: C 17.0; H 2.4; N 28.7.

### 3.1.3. Synthesis of {[Sm(btrm)$_2$(NO$_3$)$_3$]·H$_2$O}$_n$ (**3**) Coordination Polymer

Compound **3$^{Sm}$** was synthesized similarly to compound **1** using Sm(NO$_3$)$_3$·6H$_2$O (0.10 mmol, 0.045 g) and btrm (0.20 mmol, 0.030 g). Yield: 50% (0.032 g). Elemental analysis (%): Calc. for C$_{10}$H$_{14}$N$_{15}$O$_{10}$Sm: C 18.5; H 2.2; N 32.0. Found: C 18.6; H 2.2; N 30.8.

### 3.1.4. Synthesis of {[Dy(btrm)$_2$(NO$_3$)$_3$]·H$_2$O·MeCN}$_n$ (**4**) Coordination Polymer

Hot acetonitrile solution (2.0 mL) of btrm (0.10 mmol, 0.026 g) was added with stirring to a solution of Dy(NO$_3$)$_3$·5H$_2$O (0.050 mmol, 0.024 g) in acetonitrile (1.0 mL). The resulting solution was cooled to 4 °C. Upon cooling, the solution began to cloud and a precipitate was formed, which was filtered off, washed with acetonitrile, and air-dried. Yield: 87% (0.11 g). Elemental analysis (%): Calc. for C$_{12}$H$_{17}$N$_{16}$O$_{10}$Dy: C 20.4; H 2.4; N 31.7. Found: C 19.8; H 2.4; N 31.0.

### 3.1.5. Synthesis of {[Gd(btrm)$_2$(NO$_3$)$_3$]·H$_2$O}$_n$ (**5**) Coordination Polymer

Compound **5** was synthesized similarly to compound **1** using Gd(NO$_3$)$_3$·6H$_2$O (0.20 mmol, 0.091 g) and btrm (0.40 mmol, 0.060 g). Yield: 78% (0.10 g). Elemental analysis (%): Calc. for C$_{10}$H$_{16}$N$_{15}$O$_{11}$Gd: C 17.7; H 2.4; N 30.9. Found: C 17.3; H 2.1; N 29.4.

### 3.1.6. Synthesis of Single Crystal of the Coordination Polymers {[Ln(btrm)$_2$(NO$_3$)$_3$]·xMeCN}$_n$ (**1a–4a**)

Single crystals of compounds {[Ln(btrm)$_2$(NO$_3$)$_3$]·xMeCN}$_n$ were obtained by mixing acetonitrile solutions of btrm (2.0 mL, 0.40 mmol) with acetonitrile solutions of Ln(NO$_3$)$_3$·nH$_2$O (1.0 mL, 0.20 mmol, $n$ = 6 for Ln = Eu, Sm; $n$ = 5 for Ln = Tb, Dy) with stirring on a magnetic hot plate. The resulting solutions were left for slow crystallization at 4 °C.

### 3.2. *Methods of Characterization*

Elemental analysis data were obtained according to the standard procedure on a Vario MICRO cube CHNS analyzer (Elementar Analysensysteme GmbH, Langelselbold, Germany). A Scimitar FTS 2000 spectrometer (Bruker Corporation, Billerica, MA, USA) was used to record the IR spectra of solid samples in the range of 4000–400 cm$^{-1}$, the samples were prepared in the form of suspensions in Vaseline or fluorinated oil (Table S1). A NETZSCH TG 209 F1 thermal analyzer (Erich NETZSCH GmbH & Co. Holding KG, Selb, Germany) was used for thermogravimetric analysis, which was carried out in a helium atmosphere in Al$_2$O$_3$ crucibles (load 5–30 mg, heating rate 10 K·min$^{-1}$).

The polycrystalline compounds were studied by powder X-ray diffraction (PXRD) analysis on a Shimadzu XRD-7000 diffractometer, using CoKα radiation and a Ni filter. The analysis was performed at room temperature and covered a 2θ range of 5–40° with a step size of 0.03° 2θ. Each point was accumulated for 1 s to ensure accurate measurements.

A Shimadzu UV-3101PC spectrophotometer (Shimadzu Corporation, Kyoto, Japan) was used to obtain the diffuse reflectance spectra of solid samples from 240 to 520 nm

with a slit width of 5 nm. High accuracy was maintained during the wavelength axis calibration, with an error of $\pm 0.3$ nm for the UV and visible ranges, and the measurement errors associated with scattered light were minimal at 0.01%. Barium sulfide was applied as a standard. The Kubelka–Munk function was used for calculating the optical absorption of the samples in relative units by means of the diffuse reflectance spectra.

To investigate the luminescent properties of the polycrystalline compounds, an FLSP920 spectrofluorometer (Edinburg Instruments Ltd., Livingston, Great Britain) equipped with a xenon lamp was used, and all measurements were performed at room temperature. Both excitation and luminescence spectra were corrected to the sensitivity of the detector to the emission spectrum of the lamp using an internal instrument calibration curve. The photoluminescence quantum yields were determined by calculating the ratio of the integrated emission intensity of the sample and solid anthracene, which served as the standard with a known quantum yield of 0.91 [45]. The phosphorescence kinetics were recorded on a Fluorolog 3 spectrofluorometer (HORIBA Jobin Yvon SAS, Edison, NJ, USA) equipped with a cooled PC177CE-010 photon detection module with a R2658 photomultiplier. For gadolinium(III) complex **5**, the afterglow luminescence spectrum was also obtained on a Fluorolog 3 spectrofluorometer at 77 K with a flash delay of 0.15 ms and excitation at 330 nm.

The Amsterdam Density Functional (ADF) [46] program was applied for density functional theory (DFT) calculations, which were carried out on the computing cluster of the Nikolaev Institute of Inorganic Chemistry SB RAS. The parameters of the calculations were the BLYP (Becke, Lee, Yang, and Parr) density functional [47,48], the generalized gradient approximation (GGA), as well as the TZ2P all-electron basis [49].

### 3.3. Single-Crystal and Powder X-ray Diffraction Analyses

The crystal structure of polymer compounds was determined by single-crystal X-ray diffraction (XRD) analysis, which was carried out on a Bruker D8 Venture diffractometer (Bruker Corporation, Billerica, MA, USA) with a graphite monochromator, $\lambda(MoK\alpha) = 0.71073$ Å. The measurements were performed by means of the $\varphi$- and $\omega$-scanning methods at 150 K. The SADABS [50] program was used for the absorption corrections, while the SHELXL [51] program with the OLEX2 GUI [52] was necessary to solve and refine the crystal structures. Anisotropic refinement was performed for atomic thermal displacements of non-hydrogen atoms, while the positions of hydrogen atoms were found from their geometric conditions and refined by applying a rider model. Atomic displacement ellipsoids of the 1,2,4-triazole ring in compound **2** are noticeably prolate, likely due to a slight disorder. For these atoms, the anisotropy restraint (ISOR) was applied. Additionally, the Solvent Mask function was used in the OLEX2 program to remove the electron density corresponding to two acetonitrile molecules in the system. All details and crystallographic data of the structure refinements are presented in Table S2. Additional crystallographic data are available from The Cambridge Crystallographic Data Center CCDC 2,270,131–2,270,136. This data can be obtained free of charge at: https://www.ccdc.cam.ac.uk/structures/.

### 4. Conclusions

In summary, the first examples of lanthanide(III) coordination polymers based on bis(1,2,4-triazol-1-yl)methane with the general formula $\{[LnL_2(NO_3)_3]_n \cdot xMeCN\}_n$ were obtained, and their crystal structures were determined. According to the single-crystal X-ray diffraction analysis, the organic ligand serves as a bridge between the neighboring metal ions leading to the formation of 1D coordination polymer chains. The photoluminescent properties of the ligand and obtained coordination polymers were investigated in the solid state. The ligand demonstrates a blue emission with a nanosecond excited state lifetime and a quantum yield of 13%. The antenna effect is not efficient for the studied coordination polymers, and the main reason for the observed luminescence is the enhancement of f-f transition intensity under the influence of a ligand field and non-centrosymmetric

interactions. The photoluminescence quantum yields of the coordination polymers were in the range of 1–4%, and the luminescence lifetimes values were in the submillisecond range.

**Supplementary Materials:** The following supporting information can be downloaded at: https://www.mdpi.com/article/10.3390/inorganics11080317/s1, Figure S1. Selected symmetry elements in the crystal substructures of compound **1**: (a) $P2_1/n$ space group; (b) C2/m space group; Figure S2. The thermogravimetric curve for compound **1**; Figure S3. The thermogravimetric curve for compound **2**; Figure S4. The thermogravimetric curve for compound **3**; Figure S5. The thermogravimetric curve for compound 4; Figure S6. Luminescence decay plot for bis(1,2,4-triazol-1-yl)methane (btrm); Figure S7. Luminescence decay plot for compound **1**; Figure S8. Luminescence decay plot for compound **2**; Figure S9. Luminescence decay plot for compound **3**; Figure S10. Luminescence decay plot for compound **4**; Table S1. Assignment of the IR bands ($cm^{-1}$) of bis(1,2,4-triazol-1-yl)methane (btrm) and coordination polymers **1**–**4**; Table S2. Crystallographic data and structure refinement details for the coordination polymers.

**Author Contributions:** Conceptualization, E.V.L.; methodology, E.V.L.; investigation, E.A.I., K.S.S., I.P.P. and A.S.P.; writing—original draft preparation, E.A.I., K.S.S. and I.P.P.; writing—review and editing, A.S.P. and E.V.L.; visualization, E.A.I., K.S.S. and I.P.P.; supervision, E.V.L. All authors have read and agreed to the published version of the manuscript.

**Funding:** This work was supported by the Ministry of Science and Higher Education of the Russian Federation, № 121031700321-3 (The Nikolaev Institute of Inorganic Chemistry SB RAS).

**Data Availability Statement:** The data presented in this study are available on request from the corresponding authors.

**Acknowledgments:** The authors thank the XRD Facility of NIIC SB RAS for collecting the X-ray diffraction data.

**Conflicts of Interest:** The authors declare no conflict of interest.

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
