# Peer review of "Photoluminescent Lanthanide(III) Coordination Polymers with Bis(1,2,4-Triazol-1-yl)Methane Linker"

_inorganics, doi:10.3390/inorganics11080317_

Round 1

Reviewer 1 Report

This work reports the first lanthanide coordination polymers with bis(1,2,4-triazol-1-yl)methane (btrm) featuring four luminescent lanthanide ions. CPs are fully characterized, including by X-ray diffraction, thermal analysis and vibrational spectroscopy. Some luminescence data are also reported.

Generally speaking, the work is of good quality and could be reproduced without problem. The paper would be suitable for publication in Inorganics after the following remark is taken into consideration.

I do not completely concur with the explanation given for the observed f-f luminescence quantum yields. Indeed, it is common to observe f-f transition in absorption/diffuse reflectance spectra of chelates when the ligand bands are in the UV/blue (see for instance PCCP 2009, doi = 10.1039/b816131c). That the quantum yield is low does not mean necessarily that excitation goes through f-f transitions only (do the authors have a proof that they are enhanced? Have they measured their molar absorption coefficients?). The authors should measure the phosphorescence of the ligand at low temperature (77K) in the Gd coordination polymer, in order to determine the position of the triplet state. Then they could compare (a) the energy gap between the 0-component of T1 and the emitting level of the Ln(III) ion, and (ii) see if there is intensity change of this band when going to Sm, Eu, Dy. Finally, do the authors observe f-f luminescence when the chelates are excited at 280 nm?

Author Response

I do not completely concur with the explanation given for the observed f-f luminescence quantum yields. Indeed, it is common to observe f-f transition in absorption/diffuse reflectance spectra of chelates when the ligand bands are in the UV/blue (see for instance PCCP 2009, doi = 10.1039/b816131c). That the quantum yield is low does not mean necessarily that excitation goes through f-f transitions only (do the authors have a proof that they are enhanced? Have they measured their molar absorption coefficients?). The authors should measure the phosphorescence of the ligand at low temperature (77K) in the Gd coordination polymer, in order to determine the position of the triplet state. Then they could compare (a) the energy gap between the 0-component of T1 and the emitting level of the Ln(III) ion, and (ii) see if there is intensity change of this band when going to Sm, Eu, Dy. Finally, do the authors observe f-f luminescence when the chelates are excited at 280 nm?

Thank you for careful reading the manuscript and your comments. The phosphorescence of the ligand at low temperature (77K) in the Gd coordination polymer was measured as proposed by the respected Reviewer, and the position of the triplet state was estimated as 20400 cm-1 (490 nm, see Figure 7 in the main text). This position is also in agreement with our DFT calculation predicted value 21400 cm-1. So, in principal, good energy transfer should be observed in case of Sm and Eu, and no transfer is possible for Tb and Dy complexes. However, the antenna effect is improbable, since in the excitation spectra of all complexes the characteristic absorption band of the ligand with a maximum at 280 nm is absent or only weakly present in comparison with strong f-f transitions (see Figure 6 in the main text).

We have to propose another mechanism of Ln sensitization, based on enhancement of f-f transitions intensity. However, we agree with the Reviewer, that f-f transitions could be seen in the diffuse reflectance spectra of the coordination compound, and we cannot prove with complete certainty that these transitions are enhanced in our case. We also cannot measure the molar absorption coefficient of these transitions, as our samples are solid. So we keep our suggestion as a possible one, and update the discussion part of the luminescent properties (lines 195-220).

Reviewer 2 Report

The authors report the synthesis of new lanthanide compounds with bridging nitrogen ligands providing a polymeric structure, and their extensive characterization including an assessment of the luminescent properties. The manuscript is sound and well written, and I recommend its publication as is, except for two minor points: 1) page 3, section 2.2: I would specify at the beginning how the crystals were formed; 2) the charge "3+" of lanthanide ions is not always written as superscript (e.g. at line 89).

Author Response

The authors report the synthesis of new lanthanide compounds with bridging nitrogen ligands providing a polymeric structure, and their extensive characterization including an assessment of the luminescent properties. The manuscript is sound and well written, and I recommend its publication as is, except for two minor points:

1) page 3, section 2.2: I would specify at the beginning how the crystals were formed;

Thank you for your positive evaluation of our manuscript. The method for obtaining the single crystals was described in lines 107-109 of the revised manuscript.

2) the charge "3+" of lanthanide ions is not always written as superscript (e.g. at line 89).

All necessary changes have been added according to reviewer comments.

Reviewer 3 Report

Review of the manuscript entitled "Photoluminescent lanthanide(III) coordination polymers with bis(1,2,4-triazol-1-yl)methane linker".

The authors present and discuss the synthesis and characterisation of coordination polymers of europium(III), terbium(III), samarium(III) and dysprosium(III) with the ligand bis(1,2,4-triazol-1-yl)methane. The authors studied the luminescent properties of the coordination polymers.

Comments and queries

1.- The authors state: "Therefore, Ln(III) luminescence arises 20 from the enhancement of f-f transition intensity under the influence of the ligand field and non-21 centrosymmetric interaction." I want the authors to discuss what they mean about the influence of the ligand field. I agree that there is no efficient energy transfer from the ligand to the lanthanide ions.

2.- The authors characterised the compounds by X-ray crystallography, which seems reasonable. However, I could not download the crystal structures from the CCCD database. They used IR spectroscopy, thermogravimetric analysis and XRD powder diffraction analysis. I want to access the crystal structures of the polymers.

3.- They studied the polymers' luminescence and got their excitation, emission, and diffuse reflectance spectra. I agree with their conclusions about the ligand not sensitising the luminescence, and this is also clear from the lifetime data. The quantum yields of polymers are low. I would like to know if the authors could explain why the ligand does not act as an antenna for these lanthanide ions.

4.- Authors should correct some English errors.

There are several English errors throughout the manuscript. Please correct them.

Author Response

1.- The authors state: "Therefore, Ln(III) luminescence arises 20 from the enhancement of f-f transition intensity under the influence of the ligand field and non-21 centrosymmetric interaction." I want the authors to discuss what they mean about the influence of the ligand field. I agree that there is no efficient energy transfer from the ligand to the lanthanide ions.

We mean that non-symmetric ligand field leads to partial enhancement of f-f transition probability due to non-centrosymmetric interaction and possible mixing with the apparently allowed transitions due to the presence of ligands (for example, LMCT transitions). We have updated the discussion in the main text according to Reviewer’s remark (lines 195-220).

2.- The authors characterised the compounds by X-ray crystallography, which seems reasonable. However, I could not download the crystal structures from the CCCD database. They used IR spectroscopy, thermogravimetric analysis and XRD powder diffraction analysis. I want to access the crystal structures of the polymers.

For obtained structures, the CCDC numbers have been assigned and included in the manuscript (line 312). They should be available for the reviewers upon a manual request. For the convenience of the reviewers and the readers all CIF and CheckCIF files were included with the manuscript as supplementary materials.

3.- They studied the polymers' luminescence and got their excitation, emission, and diffuse reflectance spectra. I agree with their conclusions about the ligand not sensitising the luminescence, and this is also clear from the lifetime data. The quantum yields of polymers are low. I would like to know if the authors could explain why the ligand does not act as an antenna for these lanthanide ions. 

We have only tentative explanation of the absence of energy transfer in the case of our coordination polymers. Probably, the electron transfer takes place instead of the energy transfer leading to the absence of a common antenna effect. But this suggestion should be supported by additional experiments using electrochemical methods or by quantum chemistry calculation which is out of scope of the current paper (lines 195-220).

4.- Authors should correct some English errors.

English language of the manuscript has been checked and edited.

Round 2

Reviewer 1 Report

In this revised version, the authors have taken the reviewers' concerns into consideration and have modified their ms accordingly.

It, however occurs to me that I have overlook one point in my first review, regarding the excitation spectra: have they been corrected for the emission spectrum of the lamp? The emission of the Xe lamp falls dramatically down below 300 nm and this could explain why the contribution of the ligand is so low in these spectra. Please check and, in case, correct the interpretation regarding f-f transitions, in which I do not believe too much since I do not see why it should be different from other complexes.

Sorry for this overlooking.

Author Response

Yes, both excitation and luminescence spectra were corrected to sensitivity of a detector to emission spectrum of the lamp using internal instrument calibration curve. We have added this information to «Materials and Methods» Section. Moreover, it is not a question of low intensity of the lamp below 300 nm – as you can see in Figure 6 in Revision, in excitation spectrum of the ligand there is intensive band with maximum of 280 nm which is absolutely absent in excitation spectra of the complexes.

We can also give you an example of classical “antenna mechanism” from our recent publication in ICA. The same instrument was used to obtain both excitation and emission spectra. As we can see, in this case we observe characteristic bands of the ligand in excitation spectra of both ligand and Ln complexes, though only for Eu complex the energy transfer takes place.

So we are certain that in the case of btrm ligand we do not observe the energy transfer from the ligand to Ln ion, and the observed rather weak luminescence (QY less than 5% for all complexes) is due, most probably, to enhancement of f-f transitions.

Reviewer 3 Report

The authors attended to all questions and queries in their manuscript.

This manuscript presents interesting results with solid conclusions and contributes to the research area of lanthanides' coordination compounds' luminescence.

The authors improved their English considerably. It is good English.

Author Response

Thank you very much for reviewing our manuscript and giving your valuable comments